# Collaborative Learning: A Qualitative Study Exploring Factors Contributing to a Successful Tobacco Cessation Train-the-Trainer Program as a Community of Practice

**DOI:** 10.3390/ijerph19137664

**Published:** 2022-06-23

**Authors:** Isabel Martinez Leal, Jayda Martinez, Maggie Britton, Tzuan A. Chen, Virmarie Correa-Fernández, Bryce Kyburz, Vijay Nitturi, Ezemenari M. Obasi, Kelli Drenner, Teresa Williams, Kathleen Casey, Brian J. Carter, Lorraine R. Reitzel

**Affiliations:** 1Department of Psychological, Health and Learning Sciences, University of Houston, Houston, TX 77204, USA; jamart58@cougarnet.uh.edu (J.M.); mkbritto@central.uh.edu (M.B.); tchen3@central.uh.edu (T.A.C.); vcorreaf@central.uh.edu (V.C.-F.); vnitturi@central.uh.edu (V.N.); emobasi@central.uh.edu (E.M.O.); kldrenne@central.uh.edu (K.D.); bjcarte4@cougarnet.uh.edu (B.J.C.); lrreitze@central.uh.edu (L.R.R.); 2Health Research Institute, University of Houston, Houston, TX 77204, USA; 3Integral Care, Austin, TX 78703, USA; bryce.kyburz@integralcare.org (B.K.); teresa.williams@integralcare.org (T.W.); kathleen.casey@integralcare.org (K.C.)

**Keywords:** tobacco cessation, tobacco-free workplace programs, train-the-trainer, communities of practice, qualitative, behavioral and mental health disorders, program implementation

## Abstract

Individuals with behavioral health conditions account for 50% of annual smoking-related deaths, yet rarely receive tobacco dependence treatment within local mental health authorities (LMHAs). As lack of training and knowledge are key barriers to providing tobacco dependence treatment, Taking Texas Tobacco-Free (TTTF) developed an iterative, 4–6-months train-the-trainer program to embed expertise and delivery of sustained education on tobacco-free workplace policies and practices in participating centers. We explore the employee “champions’” train-the-trainer program experiences using a community of practice (CoP) model to identify key contributors to successful program implementation. Across 3 different LMHAs, we conducted semi-structured individual and group interviews online at 2 time points. We interviewed each champion twice (except for 1 champion who dropped out between measurements); pre-implementation (3 group interviews; N = 4 + 4 + 3 = 11 champions); post-implementation (7 individual interviews and 1 group interview; 7 + 3 = 10 champions). Therefore, 11 champions participated in pre- and post-implementation interviews from July 2020–May 2021. Guided by an iterative, thematic analysis and constant comparison process, we inductively coded and summarized data into themes. Five factors contributed to successful program implementation: value of peer support/feedback; building knowledge, champion confidence, and program ownership; informative curriculum, adaptable to targeted populations; staying abreast of current tobacco/nicotine research and products; and TTTF team responsiveness and practical coaching/assistance. Champions reported the TTTF train-the-trainer program was successful and identified attitudes and CoP processes that effectively built organizational capacity and expertise to sustainably address tobacco dependence. Study findings can guide other agencies in implementing sustainable tobacco-free training programs.

## 1. Introduction

While the rate of smoking among the general adult population in the United States (US) in 2020 was 13.7% [1], the lowest point ever recorded since 1965, tobacco use among individuals with behavioral health conditions, (i.e., mental health and/or substance use disorders) remains at least double that, with prevalence rates ranging from 32 to 67% [2], depending upon specific diagnosis [3]. Research indicates heavier smoking, increased nicotine dependence, lower quit rates, and intensified withdrawal symptoms when quitting for individuals diagnosed with behavioral health disorders [4]. Consequently, this population accounts for over 50% of annual smoking-related deaths [5] and is at risk of dying 25 years earlier than the general population [6,7]. Given these circumstances, researchers have designated smokers with behavioral health conditions as a tobacco use disparity group, signaling the need to address tobacco addiction within this community [8].

Another troubling trend is the evolving landscape of increasingly popular alternative nicotine delivery systems. As noted by the 2020 Surgeon General’s report [1] on smoking cessation, polytobacco use of e-cigarettes and conventional cigarettes among adult smokers is high at 49.6% according to data from the National Health Interview Survey. Polytobacco use is significantly higher among those with behavioral health issues, as indicated by a recent survey of e-cigarette use by individuals in treatment for substance use conditions in 24 treatment centers, which found that 87.1% reported dual e-cigarette and combustible cigarette use in the past month [9]. Despite the well-documented need to address tobacco dependence among this population, individuals seeking treatment at behavioral healthcare centers often do not receive services to treat their tobacco dependence [10]. A recent national evaluation from 2016 indicated [10] that in the US, only 48.9% of behavioral health treatment facilities screened for tobacco use, 37.6% offered cessation counseling, only 25.2% offered nicotine replacement therapy, and 21.5% offered non-nicotine cessation medications. Additionally, the availability of evidence-based practice guidelines for treating tobacco dependence [11,12,13], including guidelines specifically targeting those with mental health disorders [14], has not resulted in increased integration of tobacco dependence services within behavioral health settings. This is despite the fact that the development and dissemination of clinical practice guidelines has been shown to be generally effective in changing health-related practices in clinical settings [15]. These findings point to the need to directly address known implementation barriers.

The main implementation barriers hindering efforts to successfully address tobacco use (cigarettes, e-cigarettes, and all other tobacco/nicotine products) among individuals with behavioral health conditions are on the provider-level and include a lack of adequate training of providers to treat tobacco dependence [2,16,17,18,19], high provider tobacco use rates, and a host of misconceptions regarding treating tobacco and substance use and/or mental health issues simultaneously [20,21], as well as clients’ ability and desire to quit smoking. Lack of behavioral health provider training and knowledge on how to treat tobacco dependence has been reported as the most common barrier to the provision of tobacco cessation treatment [17,22,23]. A recent survey on behavioral healthcare staff attitudes and practices in client tobacco cessation services found that 37% sometimes recommended use of electronic cigarettes as a cessation aid, despite the unknown safety and effectiveness of these products [19]. Many providers still subscribe to the misconception broadcast by the tobacco industry that tobacco use is “beneficial” to the psychological wellbeing of individuals with behavioral health conditions [3,9,24]; i.e., it is required to relieve stress. Another common provider misconception is that concurrent treatment of tobacco dependence and behavioral health conditions could jeopardize substance use recovery or exacerbate mental health symptoms [25,26]. Research indicates the opposite; namely, that smoking cessation is associated with improvements in mental health such as decreasing stress and anxiety [26,27,28], as well as reductions in overall substance use, lowered risk of relapse, and abstinence from other substances. Finally, substantial research has reported that the motivation to quit smoking among individuals with behavioral health conditions is comparable to the general population’s [19,28,29,30,31]. The lower quit rates documented among those with behavioral health conditions compared to the general population, therefore, have been attributed to not being provided with the specialized care and interventions they need to successfully quit [25,32,33,34]. When these individuals have been provided with appropriate treatments and tailored interventions [35], they have successfully quit at rates comparable to the general population [36,37].

The clear lag described herein between available evidence-based treatments and their translation and adoption into effective interventions in health care service organizations—approximately 17 years in duration—has been identified as “the most serious gap in the literature” [38,39], and is recognized as a critical issue to address by multiple national health agencies [40,41,42]. The provision of adequate employee training has been identified as a core implementation component in the implementation process driving provider behavior and organizational change [43]. Training of behavioral health treatment providers addresses the primary implementation barriers to tackling smoking cessation by increasing provider knowledge and confidence in delivering tobacco dependence treatments, while rectifying the various misconceptions and attitudes about tobacco use within this population [44,45,46]. Programs for training behavioral health providers on addressing tobacco dependence among their clients have been effective in increasing the provision of tobacco cessation services [17,36,47]. However, researchers have noted that sustained implementation and integration of smoking cessation interventions in practice remain a challenge, indicating that one-time training is not sufficient to result in successful intervention implementation [16,17,48]. Rather, what is needed is an ongoing educational training program focused on sharing and enhancing knowledge to improve implementation and the delivery of tobacco cessation services. The integration of evidence into practice can be challenging, involving learning and translating explicit and tacit knowledge into applied settings [49]. This process of knowledge acquisition, and application in practice can be facilitated by collective learning, exemplified by communities of practice (CoP) [50]. A CoP is a formal or informal group of people who are engaged in a collective learning process in a shared domain of interest [51] and practice. CoPs have the potential to bridge the gap between evidence-based guidelines and real-world practice and can achieve sustainable improvements in health care service delivery [52]. One such collaborative learning group that has established the foundation for the development and emergence of a CoP is the Taking Texas Tobacco-Free (TTTF) Train-the-Trainer program [18]. The purpose of the present study was to explore champion trainees’ experiences and perspectives of the TTTF Train-the-Trainer program as an applied CoP [50], and their views on factors and processes contributing to program success.

## 2. Materials and Methods

### 2.1. TTTF Train-the-Trainer

Taking Texas Tobacco-Free (TTTF) is an academic-community partnership that implements a multicomponent tobacco-free workplace program within behavioral health and substance use treatment centers to increase the capacity for and provision of evidence-based interventions for treating any tobacco/nicotine dependence among employees and clients [53,54,55]. The TTTF program includes adoption of a 100% tobacco-free workplace policy that covers all tobacco products, including electronic nicotine delivery systems, as well as implementing tobacco use screenings that assess clients for use of these products. To date, TTTF has partnered with 23 local mental health authorities (LMHAs) in the implementation of this comprehensive tobacco-free workplace program [22,47,54,56,57,58]. LMHAs are non-profit, state-funded agencies throughout Texas responsible for providing behavioral health services through community mental health centers within a safety-net healthcare system that serves underserved, lower-income individuals [59]. While provision of general employee and specialized provider tobacco education has been a cornerstone of the TTTF program that has successfully increased provider delivery of tobacco cessation services to clients [58], the effects of the education were continuously being lost due to high employee turnover [60,61]. The TTTF Train-the-Trainer program was developed to embed sustainable, local expertise within LMHAs on the general harms of tobacco use and how to treat tobacco dependence using evidence-based tobacco cessation interventions [18]. The program consisted of training local center volunteer program “champions”—a provider or managerial member, who was not additionally compensated—to become in-house trainers in the delivery of tobacco education training to LMHA employees to ensure that expertise in evidence-based practices for tobacco management are established and not lost to employee turnover over time. Champions would serve as in-house resources for the continued training of center employees, and would train others to replace themselves, if need be.

The program consisted of training that progressed across 3 stages and was delivered live online over the course of 4–6 months from July 2020–May 2021, according to differing start and completion dates for individual LMHAs (Figure 1).

Firstly, TTTF members trained a cohort of program champions at each LMHA (3 cohorts total) on tobacco dependence and treatment among individuals with behavioral health conditions through provision of a 4–5 h comprehensive training, and then through a scaled down 90 min adaptation of the training (both trainings can be delivered “live” online, or in-person) that could serve as a model for champions of the trainings that they would later deliver to fellow employees. Secondly, champions individually delivered a “mock” 90 min training adapted to their centers’ needs at least 2 times to other program champions and to TTTF trainers who coached them and provided oral as well as written feedback and delivery tips from TTTF training observers. Thirdly, once champions had been rated as “very good” or “excellent” by TTTF training observers on evaluative items, they delivered 2 actual trainings to their fellow employees who also rated the trainers’ delivery of the training in writing and completed the same knowledge test both pre- and post-training to assess knowledge gain. Finally, champions rated their training experience with the TTTF program and the provided curriculum. Throughout this process, champions participated in the training as a cohort of 3–4 individuals, by LMHA; each champion participated in their peers’ training. Please see Nitturi et al., 2021 [18] for additional program details.

A prior study evaluating the train-the-trainer program [18] indicated successful achievement of training goals: (1) increasing self-efficacy of champions to deliver training; (2) meeting implementation fidelity on trainer effectiveness, and increasing knowledge among attending employees, as measured via employee evaluation of training; (3) stakeholder program acceptability; and (4) increasing follow-up champion-provided training, signaling program adoption. The focus of this study is on the process of training program champions.

### 2.2. Ethical Approval

All procedures for this research were approved by the Internal Review Board of the University of Houston (STUDY00002164, approval date 8 April 2020). LMHA leadership provided written consent via a memorandum of understanding prior to study participation. Researchers discussed the nature of the study and conducted interviews with each participant, who consented orally, prior to study participation. Permission was obtained from all participants for audio and video recording of interviews.

### 2.3. Study Design and Participants

An exploratory qualitative research design was adopted as most effective to explore champions’ experiences and perspectives [62] of the TTTF Train-the-Trainer program using semi-structured individual and group interviews. Participants (N = 11) were the volunteer program champions of the TTTF Train-the-Trainer program recruited from different LMHAs (N = 3) (Table 1) that had previously participated in the TTTF comprehensive tobacco-free workplace program, most of whom were providers delivering direct services to clients, and some of whom were program managers. Sample size was determined primarily by funding considerations, which stipulated active program implementation in ≥2 LMHAs. Champions were not compensated for study participation.

### 2.4. Data Collection

Total population sampling, a type of purposive sampling, was used whereby interviews were sought with the entire population of study participants [63]. Interviews were conducted by the first author, IML, a cultural anthropologist and public health researcher experienced in qualitative research, and lasted approximately 35–60 min. Semi-structured interview guides were drawn up based on research aims, were field tested and refined according to responses in the field [64], and were used to conduct individual and group interviews with program champions from July 2020 to May 2021. Group interviews consisted of 3–4 champions. Given safety concerns regarding COVID-19 transmission, all interviews were conducted online and recorded using a videoconferencing platform [65].

Individual and group interviews with champions were conducted at 2 time points, pre- and post-implementation of the program. Such that, with one exception described below, each champion was interviewed twice. A pre/post design was adopted to allow for a formative evaluation process in which data collected pre-implementation were used to understand and adapt the program to clinic-specific implementation contexts and needs prior to implementation. Pre-implementation interview questions focused on current trainings (annual, in-service, new employee) offered on tobacco use, and treatment, the feasibility of providing additional tobacco education training, the ideal structure (length, format), curriculum content, and delivery method of such trainings, and potential implementation barriers. Post-implementation interview questions focused on champions’ experiences and views of the training, what was positive and negative about the training, suggested improvements, how the training affected confidence in delivering tobacco training in their organization, views on the training structure, curriculum, and materials, and additional suggestions or supports needed to effectively deliver training to fellow employees.

### 2.5. Data Analysis

All interview recordings were transcribed verbatim by a professional transcription service and uploaded onto Atlas.ti 9 (Atlas.ti, Scientific Software Development, version 9.1.6, Berlin, Germany, 2020) to organize and facilitate data analysis. Data were initially analyzed using thematic analysis and constant comparison to identify and refine themes within the data [66]. During data interpretation, a CoP framework was subsequently applied to further guide final analysis [50], as described below.

Coding entailed an iterative process, using constant comparison to continuously compare new and emerging data to previously coded transcripts to condense codes into categories and themes. Two members of the research team, IML and JM, the first, and second authors, respectively, both trained in qualitative analysis, initially independently coded the first 5 transcripts, and then met to compare and discuss coding, and to develop an initial codebook that was reapplied to all the data. The codebook remained open to refinement, with new codes being added as needed, until analysis of successive interviews no longer yielded any new codes [67]. Each champion was interviewed twice (with the exception of a champion who dropped out between measurements, noted below), both pre- and post-implementation, which sufficed to attain repetition of codes across transcripts during data analysis.

### 2.6. Theoretical Framework: Communities of Practice

A CoP model was used to frame champions’ experiences of the TTTF Train-the-Trainer program, and to explore the factors contributing to the process of successful knowledge, and practice enhancement. CoPs are groups of people engaging in collective learning who interact regularly about a shared domain of interest [50]. While similar to other professional or collaborative networks, CoPs are distinct in focusing on knowledge sharing, learning, and creation [68]. CoPs can vary in whether they are informally self-organized or formally organized groups, in size, as well as how they interact, e.g., virtually or in-person. Originally it was developed by Lave and Wenger in 1991 as a theory of “situated learning” that stressed how learning is a complex social process involving working, learning, and innovating, best described as “learning-in-working [69]”. This conception views learning (knowledge), and working (practice) as integral to each other, rather than as being in conflict. CoPs represent the “fluid evolution of learning through practice”, focusing on how learning forms and changes and innovates work communities [69]. There are 3 crucial characteristics to CoPs: (1) *domain*: a shared domain of interest that identifies the community and its members; (2) *community*: members who pursue their interest in the domain, engage in joint activities, and share information; and (3) *practice*: the development and sharing of a repertoire of resources or practices [51]. Taken together, these characteristics allow CoPs to be adaptive and responsive to changing circumstances and sources of innovation and change in the workplace, as well as drivers of implementation of evidence-based interventions. We explored champions’ experiences of the TTTF Train-the-Trainer program as a CoP that focused on supporting the implementation of evidence-based tobacco cessation knowledge and practice within behavioral health treatment settings. While the training program was not explicitly set up as a CoP, CoPs can be informal groups collaborating in learning and sharing knowledge; the training learning groups established through the program functioned as CoPs.

## 3. Results

Qualitative interviews were conducted at 2 time points online, pre- and post-implementation with the same champions. Pre-implementation, we conducted 1 group interview at each of the 3 different LMHAs, which included 4 champions at LMHA1, 4 champions at LMHA2, and 3 champions in LMHA3, for a total of 11 champions participating in group interviews. One champion dropped out of the training program prior to post-implementation procedures due to competing work demands, leaving us with 10 remaining champions. Post-implementation, with the same 10 champions that remained, we conducted 7 individual interviews (1 champion at LMHA1, 3 at LMHA2, and 3 at LMHA3) and 1 group interview (3 champions at LMHA1). Eleven champions engaged in pre-implementation group interviews, and 10 across pre- and post-implementation individual interviews and group interviews, with 11 total participants engaging in interviews altogether across the two time points between July 2020 and May 2021.

Data analysis yielded 5 themes on champions’ experiences of the train-the-trainer program and their perceptions of factors contributing to successful program implementation: (1) collaborative learning: value of peer support, practice, and feedback; (2) building knowledge, increased champion confidence, and program ownership; (3) informative curriculum, adaptable to targeted populations; (4) staying abreast of changing tobacco/nicotine research and evidence-based practice; and (5) facilitated practice: responsiveness and practical coaching/assistance by TTTF team. Pseudonyms are used in reporting quotes.

### 3.1. Collaborative Learning: Value of Peer Support, Feedback, and Practice

#### 3.1.1. Value of Peer Support

Champions highly valued the opportunity to learn collaboratively afforded by the peer training model of the program. This “learning together” not only allowed champion trainees to deepen their own knowledge of the educational material, but also to practice and refine their own training skills and to engage in the provision and receipt of support from peer trainees. Group learning also allowed champions to reflect upon their own training and delivery skills for improvement through observing how peers presented the material and developed their presentation styles. The training program functioned as an apprenticeship for champions, where they apprenticed and learned from their peers, as well as the TTTF trainers. The peer training model of the program was valued as particularly beneficial by champions:


*I agree with Sheila, she’s pointed out several times that we all learned a lot from each other in sitting in on each other’s trainings as well. So really, we sat through maybe 10 trainings. We all fed off that, and we all gained bits and pieces of everybody else’s techniques as well as their points of interest that they pointed out maybe better than we did. So I think it was a huge benefit for us to sit in each other’s trainings.*
(Lisa, LMHA1)

#### 3.1.2. Feedback

Feedback from fellow employees receiving the tobacco education trainings was also valued by champion trainees. In addition to receiving written evaluations from employee attendees as part of the TTTF training program, many program champions also reported that attendees also informally communicated with them verbally on the effectiveness of their delivery as well as the content of the training:

*One of* [my coworkers] *came up to me and told me about this last* [employee training], *“My God, I was so impressed by the training”, and that helps me to get better. …*
*It’s always good to get feedback from people. Especially people who work with you because they’re going to be honest.*(Veronica, LMHA1)

#### 3.1.3. Practice

Practice—in the form of practice training sessions—was also recognized as important to the learning process. These practice sessions allowed for review and perfecting of champions’ delivery of the educational material, and allowed them to clarify any gaps in their own knowledge:


*It would’ve not been helpful at all if we had not had those practice sessions with each other and with you guys. If we had gone from you giving us the training straight into doing training for our staff, I don’t think it would’ve gone as well… especially where I think I’ve got holes I find myself less confident. So, I was really grateful to have those trainings that we did with each other.*
(Sheila, LMHA1)

### 3.2. Building Knowledge, Increased Champion Confidence, Program Ownership, and Innovation

#### 3.2.1. Building Knowledge

Crucial to the process of building champions’ knowledge about tobacco use and dependence among those with mental health needs was valuing the information they were receiving and the expertise of those delivering the training. While champions appreciated being provided with explicit, abstract knowledge in the training process, they more highly valued the transmission of implicit knowledge, learned in the practice of treating tobacco use:

*You guys are so very knowledgeable, you were an excellent resource, and knew the material front to back. That helped us feel more confident because you guys were so at ease with the material, and I appreciate that…You were like—‘Hey, this is what works, this is what we’ve seen that doesn’t work’…**I felt that you were there to support us from the beginning and being really approachable and just normal people, trying to help people learn about tobacco use. So I think that was a big part of our learning. We didn’t feel threatened or intimidated in any way to try to do it perfect. Because I think you guys said from the beginning, we’re all going to have different styles, we’re all going to have different parts of that presentation that we do**better, and some places that we’re not as good presenting. And like S.* [peer] *said, I think we fed off of that we were able to see different things through being able to learn from everybody else’s presentation.*(Lisa, LMHA1)

#### 3.2.2. Increased Champion Confidence

As noted above, equally important to transferring and enhancing knowledge was creating a learning environment in which champions felt comfortable with trainers, and that they were approachable and helpful rather than critical and intimidating. The same champion continued to report how provision of a supportive learning community made up of trainers and peers was crucial to fostering champion confidence:


*You all were very patient and the guidance that you gave us was good. In the beginning, I was a little bit confused… but once you all started guiding me… you slowed it down, and I think giving it to us in pieces really worked and helped my self-confidence to be able to do the presentation.*
(Clara, LMHA2)

#### 3.2.3. Program Ownership: Becoming a Tobacco Treatment Specialist

Through building knowledge and confidence in themselves as trainers, champions also addressed developing program ownership through a process of making the training “their own”, which entailed enrichening the training with their own personal stories about how to apply the abstract knowledge presented in working with clients:


*As I’m going, I think of stories I can say along the way. It reminds me of stuff I’ve experienced in the past with clients that I’m able to add little stories to for these different topics, to help people quit smoking. It was challenging, but then with the support and assistance and the help that you all gave me and the encouragement, I was able to open up, and I was able to make it a little bit of my own, and the more I did it the more excited I would get, and then I got inspired and so I was able to do the presentation with a flow and to make it more of my own.*
(Clara, LMHA2)

Through this collaborative learning process, in which champions built confidence and program ownership, they also developed the competence and expertise to become and to see themselves as tobacco treatment specialists:


*I think it was a very good training because—I was kind of nervous at the beginning because there was a lot of information and learning, but at the end, it was all worth it, I feel confident. … This program was fairly new to me, and if you would’ve asked me in the beginning, I probably would’ve said no, I’m not confident at all to do the training… But now I feel confident. I know that I’m probably going to keep learning, and I am probably going to run into something that I might not know, but that’s a process, that’s what goes with it. I’m just happy I went through it.*
(Veronica, LMHA2)

#### 3.2.4. Innovation

Additionally, champions proposed to explore alternative approaches to training, using more innovative models such as collaborative team training:

*The other thing that we haven’t really explored, that we want to, is the training together. Like, for example, I do half the class and then L.* [peer] *finishes it up, or vice-versa. So the audience gets a mixture of different opinions and people.*(Sheila, LMHA1)

### 3.3. Informative Curriculum, Adaptable to Targeted Populations

#### 3.3.1. Informative Curriculum

Champions reported valuing the training curriculum as very informative, and that they learned a lot of information. Although some champions initially felt overwhelmed by the amount of information they received, as they progressed in the training and mastered the information, they became more comfortable and valued the comprehensiveness of the material:

*At first, it was kind of difficult… It was challenging, but with all the support, and all the important information was there—the dangers, what parts of the body it* [smoking] *affects, and the way it affects it, and it went into a nice flow into how to try to help people to quit smoking, and the different medications that they can use to stop. All this information is very helpful because when I am talking to people about tobacco, I’m able to explain to them better because I know a lot more now with this training… My co-workers said they enjoyed it, and they learned a lot. They said that everything was very informational and very educational for them. They learned stuff that they didn’t know about tobacco.*(Clara, LMHA2)

Even champions who had previously received training and certification as tobacco treatment specialists stated that they had learned a lot of new information during this collaborative learning process, and found the training to be complete:


*I know that you were providing us with a lot of new information especially given to me when I had previous knowledge. I did learn a lot. That’s what I would say, there’s nothing to improve. I know there’s always going to be more information coming in but your whole presentation, all the training you gave us, those were perfect*
*…*
*I learned a lot of information that can benefit my patients, especially how they can obtain more services… like what I learned from peers besides all the information is mainly the benefits the patients can obtain from us.*
(Juan, LMHA2)

#### 3.3.2. Curriculum Adaptable to Target Populations

Champions particularly appreciated that the curriculum could be adapted to targeted populations served by LMHAs. The training focused primarily on treating tobacco dependence among underserved individuals with mental and/or substance use disorders. However, as tobacco use and dependence are most prevalent among populations experiencing economic and social disadvantage, champion trainees were also provided some information on tobacco dependence within other vulnerable populations, including: individuals with intellectual and developmental disabilities (IDD), those with opioid use disorders, those experiencing homelessness, members of sexual/gender minorities, racially minoritized groups, as well as youth:


*I think everything about that training was good, I liked it. It’s very informative. It’s a lot of information, but it’s good. It focuses on a lot of topics, a lot of different populations…*
*I had some coworkers that work with IDD, most of their patients smoke, and they liked that information so they can also explain it to them. So, there was nothing to be added. Everybody thought it was very educational and informational.*
(Clara, LMHA2)

The capacity to tailor tobacco trainings to the needs of specific special populations was essential to most program champions. Many asked for—and received—additional information and presentations developed by TTTF that targeted tobacco dependence among those experiencing homelessness, sexual minorities, and those with opioid use disorders. Even so, champions asked for additional resources and guidance in tailoring their trainings to the needs of the special populations they served:

*I’d really like to look a little bit more at those marginalized cultures, looking at our homeless, LGBTQ+. Keeping us up-to-date on the new information that comes out, we’d love to have that… There’s just not enough of a focus* [on special populations], *so what we’d need if we were to specialize a 30-, 45-min training for a specific group, we might want to focus on the population that you work with most, and looking at a bit more data on that.*(Sarah, LMHA3)

Additionally, champions felt confident in adapting the training content to fit the needs of the particular population that they were serving:

*There were gaps* [in information] *with the IDD, I think we need to keep up with it and see what other information is available, for that and the homeless. But of course, that in general has very limited data. I just did the best I could to add more to* [the training].(Veronica, LMHA2)

### 3.4. Staying Abreast of Changing Tobacco/Nicotine Research and Evidence-Based Practice: Clarification and Additional Resources

#### 3.4.1. Staying Abreast of Changing Tobacco/Nicotine Research: Clarifications

Given the ever-changing landscape of combustible tobacco, vaping, oral tobacco, and synthetic nicotine products, champions especially appreciated the information and research provided within the training program on the latest issues, trends, and policies on these emerging products. The information provided in the training aided them in clarifying misconceptions that have been widely disseminated by the tobacco industry regarding these products. For example, presenting employee attendees with an accurate understanding of vaping devices and their harms:

*I think when we first did our* [TTTF] *training, it* [vaping] *had just started, and a lot of people were thinking it would be a good alternative to cigarettes… I’m really excited to learn more about e-cigarettes. I was surprised this morning when I taught… a lot of them thought that vaping was a NRT* [Nicotine Replacement Therapy], *and they get that confused, that they think that e-cigarettes are an alternative… A safe alternative to smoking.*(Sheila, LMHA1)

#### 3.4.2. Additional Resources

More than with any other topic covered in the training, many champions were interested in learning more about these emerging tobacco/nicotine products, and requested that TTTF team members continue sending additional information and resources on them to stay informed about developing research, policies, and products:


*You all had the different types of e-cigarettes, and like the different types of tobacco, and the chew tobacco, there’s all different types of tobacco. We try to keep up to date on all the different changing e-cigarettes because there’s always so many new ones coming out on the market all the time. Yes, because research, it changes all the time.*
*There’s always new information out there. Of course, we’d like new research also about e-cigarettes, and SNUS—since we still don’t know a lot about that. We’re still learning since it’s new.*
(Clara, LMHA2)

Champions reported being interested in receiving more information on these alternative nicotine products because of their growing popularity and use, particularly among teens and youth, to stay current with research and practices within the community:


*Especially the part that I really enjoyed and learn the most from—even though there’s still a lot for us to find out—is the part about the electronic cigarettes, the e-cigarettes. I think it’s really fascinating. It’s interesting because that’s the most popular method now especially for teens and such. I knew pretty much nothing about the e-cigarettes, and so it was good just to learn the little bit that we, that the community does know now.*
(Michael, LMHA1)

#### 3.4.3. Staying Abreast of Tobacco Cessation Evidence-Based Practices

Moreover, champions sought to stay informed about developments in evidence-based tobacco cessation research and interventions to inform their practice, and would regularly be sent webinars on these topics by the TTTF team, which they attended and reported greatly appreciating:

*I heard this morning* [on a webinar] *that they felt that ‘cessation’ wasn’t the adequate term any longer, that we needed to use the word ‘treatment’, or ‘smoking recovery’ instead of ‘cessation’, because they felt that was a much stronger word to be using to show that it is that type of addiction that’s going to require treatments for people to stop.*(Sheila, LMHA1)

### 3.5. Facilitated Practice: Responsiveness and Practical Coaching/Assistance by TTTF Team

#### 3.5.1. Responsiveness

All champions greatly appreciated the practical coaching and guidance provided by the TTTF team as facilitators of the collective learning and training process. Champions reported being grateful to the TTTF team for being available and responsive to address any of their training concerns and needs in serving as training mentors engaged in tutoring and developing champions as tobacco treatment specialists.

*And also, I want to say that B.* [TTTF trainer] *was very responsive to our questions, I sent him a lot of questions and I know that S.* [peer] *did also. But he was extremely responsive to them, and that put us more at ease too knowing that he was there if we were having a panic moment, or we were confused about something. And so, I really appreciate that.*(Lisa, LMHA1)

#### 3.5.2. Practical Coaching/Assistance

Champions spoke of the mentoring and coaching support provided by the TTTF team as valuable not only in terms of the learning and exchange of knowledge, but also as inspiring them in the value of training others in addressing tobacco use:


*It was a good experience, this is very good for people to learn and be able to educate other people about the dangers of tobacco because tobacco has been out there for a long time, and it does so much damage to people… You all did excellent in guiding us, and with the support and the assistance, you encouraged us very well and the support was really good, and you all inspired us very well, like, you want to do something with it.*
(Clara, LMHA2)

#### 3.5.3. Structure and Model

Champions also reported that the structure of the training program, in terms of the duration and use of a peer model, facilitated their learning of comprehensive material as well as practice:


*You guys have done a really great job with this training. I don’t know that there’s much more information that you could give somebody… I feel very confident in my ability to train this course… There was a lot of information to cover, and a lot of times, it seems like people try to fit a training into too short of amount of time, and then expect you to be able to regurgitate that whenever you are training somebody else. That doesn’t always work, so I loved how long our initial training was and then the fact that we had to train a couple more times in order to be certified… And not being the only one from our organization to go through this training, it was very nice for us to be able to do it as a team… In the two employee trainings that I did, I got really great feedback. People really appreciated that information. I had the opportunity to take in all that information and learn it really well… of all the train the trainer trainings that I’ve been through, this one was by far the best.*
(Jane, LMHA3)

## 4. Discussion

This qualitative study used a CoP framework to understand champions’ experiences and perspectives of the TTTF Train-the-Trainer training as a collaborative learning program and identified the key factors contributing to program effectiveness. Champions who completed the TTTF Train-the-Trainer program reported that the training was successful in building champion knowledge, increasing champion confidence to deliver the training, developing effective trainers, increasing knowledge among employee attendees of training, having the curriculum and training valued as informative and acceptable by attendees, and increasing the delivery of trainings within the LMHAs. Together, these program achievements attest to effectively building organizational capacity to treat tobacco use and dependence within these LMHAs, as training providers on treating tobacco dependence among clients has been shown to increase the provision of tobacco cessation services [17,18,47]. Below we discuss how various program factors characteristic of CoPs contributed to successful program implementation.

### 4.1. Collaborative Learning Builds Knowledge, Champion Confidence, and Professional Identity

In keeping with prior research on the benefits of collaborative learning of CoPs [69,70,71,72], champions stressed the value of a peer-based training model. Findings demonstrated that such a model allowed for increased learning through knowledge sharing between champion trainees, as well as knowledge transfer between TTTF trainer and peer trainees. Research studies confirm the effectiveness of using learning processes that employ experts to lead and facilitate peer learning [73,74], which particularly, in this case, produces collective knowledge that is then disseminated to the larger community [50,51].

Study findings attest to how using the CoP concept as a learning process promotes meaningful and collective learning through knowledge sharing by community members, who as a team together advance their own acquisition of skills and knowledge by engaging in each other’s training processes. This model also provides a novel and constructive approach to training that facilitates learning-in-working [69] founded on “collaborative working and the use of collective intelligence” [71,75]. Findings align with studies affirming a collaborative approach to learning, which underscore that learning, or knowledge transfer and exchange, is a social and dynamic process [76], and professional networks such as CoPs can play a key role in hindering or facilitating the process of bringing research into practice [77,78,79].

Champions reported especially valuing learning implicit or tacit knowledge—practical knowledge, insight, skills—that was gained through experience, rather than through formal or codified transmission [80]. They highly valued learning “what works and doesn’t work”, rather than learning abstractions that were detached from practice. Tacit knowledge was highlighted within the CoP learning process, as knowledge acquisition was a negotiation among peers of meaning in practice, which entailed shared concerns and joint learning, support, and practice [70]. Champions reported that within the context of this supportive community, they were provided with a comfortable learning environment in which their ability to learn was potentiated and their confidence was bolstered. The responsiveness, practical coaching, and assistance of the TTTF team was also reported as a key factor contributing to the success of the training program. As noted in other research, the peer model and CoP process used in the training allowed TTTF experts and champion apprentices to become familiar with each other in a more reciprocal process, facilitating greater understanding of one another’s needs, aims, preferences, and circumstances [81], which in turn enabled greater responsiveness. As the development of this tacit knowledge arises from reflective practice, the exchanging of this type of knowledge through sharing of life experiences and shared practices enhances attendee engagement and learning during trainings and the dissemination of translational and actionable knowledge [82].

Findings also show that increased champion confidence was related to program ownership in two related senses—first, in making the training “their own”, and second, in becoming or developing an identity as a tobacco treatment specialist. In personalizing the training presentation, champions demonstrate confidence which is vital to sharing relevant tacit knowledge with attendees; a knowing-in-practice that serves to engage the audience and communicates comfort and expertise with the material [83]. Other studies report the potential of CoPs to increase confidence to perform or deliver evidence-based practices [72,81], highlighting the importance and need for more training processes/models that focus on tacit knowledge. Given the development of a “safe” place [84] among peers and trainers, champions reported developing a common sense of identity, as well as identifying themselves as tobacco treatment specialists, as they together gained competencies in learning and practice that included experimenting with more innovative training models. This identification also enables sustainability of the training program.

### 4.2. Responding to Diverse Populations and Evolving Tobacco/Nicotine Landscape

The curriculum [85] was appraised as being comprehensive, and for some champions, it was overwhelming at first. However, all champions mastered the material and came to appreciate its breadth. The gaps or weaknesses regarding the curriculum delivered by champions pertained to insufficient information on e-cigarettes and on two specialized populations: sexual/gender minorities, and those who are experiencing homelessness. Champions particularly appreciated that the presentation was adaptable and could be tailored to specific special populations. Additionally, they adapted the presentations themselves, i.e., conducting their own research and adding missing information as needed, which again signals confidence in mastering the material and comfort with their abilities as trainers. The learning community model also lends itself to adaptation of materials, strategies, and delivery systems, in dialogue with community members through sharing of expertise [69].

It is significant that champions reported that the training lacked enough information on e-cigarettes and requested to be sent additional research and to be kept up-to-date on the constantly changing landscape of e-cigarettes and other alternatives to combustible tobacco and policy decisions regarding these products. Given their relative newness, available evidence on the risks of long-term e-cigarette use remains unknown. As such, within the community of tobacco researchers, there is a lack of consensus on the use of e-cigarettes in public health [86]. While there is no doubt that these products are unsafe and contain known carcinogens they apparently are less harmful than combustible cigarettes. As such, some researchers suggest that they could be a safer alternative to smoking combustible cigarettes, acting akin to nicotine replacement therapy as an aid to quitting smoking—a marketing pitch promoted by the tobacco industry manufacturing these products. Others point to research indicating that most e-cigarette users end up with dual usage—smoking combustible and e-cigarettes, thus defeating the use of e-cigarettes as a quit aid [87]. Following the clinical guidelines of the U.S. Preventive Services Task Force [13], the TTTF program does not promote the use of e-cigarettes as a quit aid but recommends the use of FDA-approved nicotine replacement therapy. In the midst of this ongoing debate, champions followed the guidelines recommended by TTTF and clarified to training attendees that e-cigarettes are not a recommended and safe nicotine replacement therapy but are themselves harmful. Champions asked that TTTF keep them informed about this debate and provide them with additional resources on alternative nicotine products as they were keen to educate their fellow employees and clients on evidence-based practices to assist them to quit smoking and address any misconceptions concerning these products. As a host of provider misconceptions regarding tobacco use among individuals with behavioral health conditions are cited as key barriers to treating tobacco use among this population, clarifying these and other misconceptions on alternative nicotine products is important for providers in implementing evidence-based practice. Through having been trained as tobacco treatment specialists themselves, champions were aware of the importance of employee training to increase the provision of tobacco interventions to individuals with behavioral health conditions [17,36,47], and sought to be diligent in their responsibilities as trainers. Furthermore, provision of employee training has been recognized as a key driver of changing provider behavior within implementation research [43]. This program successfully trained champions to become trainers themselves, and supported them in interpreting, adapting, and building upon the training.

### 4.3. Strengths, Limitations, and Future Research

The adaptability of the TTTF Train-the-Trainer program to the needs of individual centers and to diverse underserved populations is a strength of the program. Another strength is the development of a step-by-step implementation guide that includes all training materials—the curriculum, training slides and videos, and evaluation materials—as part of this project, which are freely available online through the TTTF website for passive dissemination to encourage program implementation by interested behavioral health centers [84]. While this study underscores the crucial role played by expert trainers/facilitators in the collective learning process of training program champions, the detailed instructions and materials provided allow other interested health care treatment facilities to replicate program implementation using their own employees. Thus, health care treatment facilities with limited financial resources could implement the TTTF Train-the-Trainer program partially or wholly. In applying a CoP process to understand the factors contributing to successful implementation of the program, this study expands upon the train-the-trainer literature by supporting the establishment of sustained communities of practice and training. Limited follow-up with participating LMHAs indicates sustained training of employees by champions following program completion.

While we purposively recruited LMHAs that had been previous TTTF program partners, as they were already familiar with the comprehensive TTTF program, this also presents a limitation. Future research should evaluate the implementation and outcomes of this program in health settings that have not previously implemented the TTTF program to assess the feasibility and adoption of the training. Additionally, findings may not be transferable, given the primacy of social dynamics and interactions among members of this learning community in regards to effective program implementation. Future mixed methods studies should be undertaken to explore the conditions that facilitate the development of collective learning within communities of behavioral health providers who provide tobacco cessation services to support the implementation of evidence-based practices.

## 5. Conclusions

The TTTF Train-the-Trainer program serves as an effective model for implementing a sustained training program that is superior to one-time trainings [16,48], shown to be insufficient in successful tobacco cessation intervention implementation. Champions attributed program effectiveness largely to factors related to training within a learning community. Study findings indicate that the establishment of a TTTF Train-the-Trainer learning community facilitated a continuing training program that can enhance the promotion and delivery of evidence-based tobacco treatment within behavioral health settings. The establishment of such learning communities also better equips health care centers to respond to the high employee turnover common within these settings [60]. Adoption of the TTTF Train-the-Trainer program transformed the delivery and training of evidence-based practices from transference of explicit, decontextualized, abstract knowledge into a sharing between experts and apprentices of tacit knowledge that was grounded in the complexities of practice experienced within the community of practitioners. This describes a contextualized model of learning that promotes a knowing-in-practice and translational results; where knowledge transfer is situated and adapted to the context in which it is developed and used [81]. Thus, evidence is integrated into practice. In doing so, it lends itself to facilitating the implementation of evidence-based practice, effectively building organizational capacity and expertise to sustainably address tobacco dependence within healthcare settings. Moreover, successful implementation of the train-the-trainer program facilitated provider and organizational change in treating tobacco dependence, as provision of adequate employee training is recognized as a core implementation component that drives provider behavior and organizational change in the implementation process.

## Figures and Tables

**Figure 1 ijerph-19-07664-f001:**
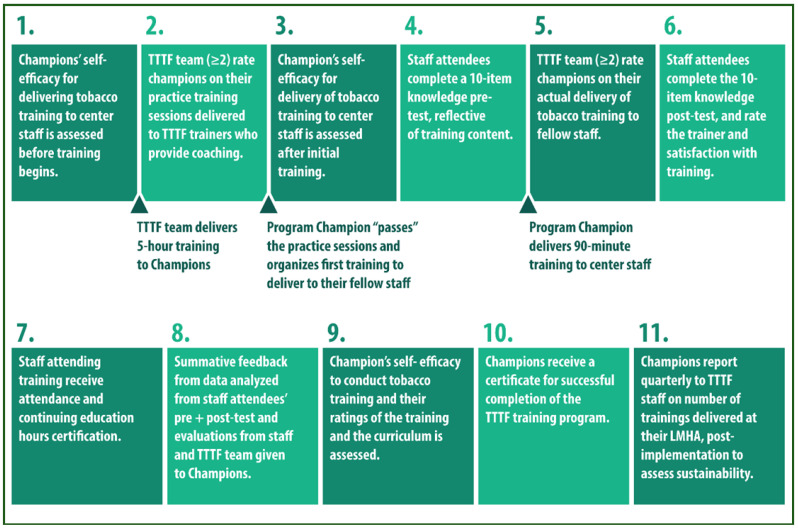
Infographic summarizing the steps involved in implementing the Taking Texas Tobacco-Free (TTTF) “Train-the-Trainer” program. Main events are marked underneath in between some steps. Reprinted/adapted with permission from Ref [18]. 2021, L. R. Reitzel on behalf of all coauthors.

**Table 1 ijerph-19-07664-t001:** Characteristics of participating centers.

Center	Number of Individual Clinics	Number of Full-Time Employees	Number of Full-Time Providers	Total Annual Clients Served	Total Annual Unique Client Contacts	Counties Served (%Rural)
LMHA1	42	247	150	92,498	5420	23 (100%)
LMHA2	20	323	254	229,482	9808	4 (50%)
LMHA3	31	419	286	239,672	11,243	6 (88.3%)

Note: LMHA = local mental health authority.

## Data Availability

The data presented in this study are available on request from the corresponding author. The data are not publicly available due to privacy and confidentiality concerns given the very small group of local mental health authorities and their participating champions, facilitating the ability to link champions with their data, which could affect dynamics of ongoing work relationships in unknown ways.

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
