# Peer review of "Collaborative Learning: A Qualitative Study Exploring Factors Contributing to a Successful Tobacco Cessation Train-the-Trainer Program as a Community of Practice"

_ijerph, 2022, doi:10.3390/ijerph19137664_

Round 1

Reviewer 1 Report

Summary
This manuscript reports on a rigorous, qualitative evaluation of a train-the-trainer model of smoking cessation/treatment referral and provision in Texas, within the context of a Community of Practice (TTTF). The manuscript is well written and organized well. Minor comments and recommendations are outlined for each section. The only moderate criticism concerns the significant overlap between details in the Results and rehashing them in the Discussion. Greater effort could be put forth to trim the Discussion and discuss how this work relates to clinical implementation and future research.

Title
The title is appropriate and informative.

Abstract
1) “Behavioral health clients” is an ambiguous term and person-first language is recommended. Please consider rephrasing throughout. “Individuals with behavioral health conditions” (Intro) or similar is preferable.
2) The sample associated with “(N=21)” is unclear. Does this mean 21 individuals participated across the 4 online interviews?

Introduction
1) Lines 72-74: the language here suggests strong causation. Are these findings correlational (e.g.,individuals able to quit have lower stress and anxiety)? Or were these reported through RCTs?

Methods
1) Section 2.1: (very minor) rather than redefine them as “participants,” why not stick with “champions” throughout? This avoids potential confusion for readers who miss or skip this line.
2) Section 2.2: the formatting of Figure 1 (in between stage 1-6 and 7-11) gets a bit wonky and was hard to evaluate.
3) Section 2.3: table 1 has some formatting inconsistencies and spacing typos (e.g., final column).
4) Section 2.5: how many successive interviews were needed that did not generate new codes before additional coding was stopped?

Results
1) Overall, the results are extremely well written – selecting relevant quotations to illustrate the themes.
The only criticism concerns the relatively long length of this section. Fewer and/or abbreviated quotes may help reduce the length.
2) The sample is still unclear as described in the opening sentence. Instead of “at 3 different” would “across 3 different” be accurate. “At” makes it sound like you got 10 champions from each site and 30 would be the expected total. Is it clear later about the distribution from each site? And the distribution by type of interview?

Discussion/Conclusions
The Discussion is also well written but highly repetitive with the results (and organized in an identical manner).
It’s also a bit long and the major findings could be summarized more succinctly.

Good work and good luck!

Reviewer 2 Report

Collaborative learning: A qualitative study exploring factors contributing to a successful tobacco cessation train-the-trainer program as a community of practice

Thank you for the opportunity to review this manuscript, which examined champion trainees’ experiences and perspectives of a train-the-trainer program focused on treating tobacco dependence. I found the manuscript to be extremely well-written and to be of overall high quality. I have a few minor suggestions to strengthen the manuscript.

1.       It would be helpful to mention information regarding the timing of the study (e.g., in which year the training was implemented, the duration of the training) in the abstract. It would also be helpful to mention the year in which the study was conducted in the methods.

2.       When referring to smoking/tobacco use in the introduction, are the authors referring to research on cigarette use or any tobacco use? It would be helpful to be more specific early on to orient the reader.

3.       The introduction would be strengthened by providing some context surrounding the evolving tobacco landscape, including increased use of alternative tobacco products and polytobacco use, especially as this comes up as a theme in the results.

Round 2

Reviewer 1 Report

Summary
This revised manuscript reports on a rigorous, qualitative evaluation of a train-the-trainer model of smoking cessation/treatment referral and provision in Texas, within the context of a Community of Practice (TTTF). The authors took care to quickly respond to reviewers’ concerns and the manuscript is markedly improved. Enthusiasm continues to be high for this work. One minor issues remain: (in the Abstract, Methods, Results), rather than characterize the sample size as 21, it would be more appropriate to characterize it as 11 and clearly describe the pre-/post-implementation interviews. As written, particularly in the abstract, it is not apparent that 10 of the 11 post-implementation interviewees were the same champions that participated in the pre-implementation interviews. It’s easier to discern the design in the Results but still potentially misleading in the current manuscript draft. Please consider revising where necessary. Also, Figure 1 looks great! Minor note: define “TTTF” in the caption so the Figure can “stand alone.”
Good work and good luck!
